# Estimating the effect of anticipated depression treatment-related stigma on depression remission among people with noncommunicable diseases and depressive symptoms in Malawi

Josée M. Dussault[1]*, Chifundo Zimba[2], Harriet Akello[2], Melissa Stockton[3,4], Sherika Hill[5], Allison E. Aiello[1,6], Alexander Keil[1], Bradley N. Gaynes[1,7], Michael Udedi[8], Brian W. Pence[1]

1 Department of Epidemiology, UNC Gillings School of Global Public Health, Chapel Hill, NC, United States of America, 2 UNC Project Malawi, Lilongwe, Malawi, 3 Department of Psychiatry, Columbia University Vagelos College of Physicians and Surgeons, New York, New York, United States of America, 4 New York State Psychiatric Institute, New York, New York, United States of America, 5 Center for Child and Family Policy, Duke University, Durham, NC, United States of America, 6 Department of Epidemiology and Robert N. Butler Columbia Aging Center, Columbia University Mailman School of Public Health, New York, New York, United States of America, 7 Department of Psychiatry, UNC School of Medicine, Chapel Hill, NC, United States of America, 8 NCDs & Mental Health Unit, Ministry of Health Malawi, Lilongwe, Malawi

* josee.dussault@unc.edu

**Data Availability Statement:** De-identified patient-level data from the SHARP study are available via the National Institute of Mental Health Data Archive

## Abstract

### Purpose

While mental health stigma research is sparse in Malawi, research in other settings suggests that stigma represents a barrier to mental health treatment and recovery. Accordingly, we conducted an analysis to understand the role of treatment-related stigma in depression care in Malawi by estimating the effect of patients' baseline anticipated treatment-related stigma on their 3-month probability of depression remission when newly identified with depression.

### Methods

We conducted depression screening and treatment at 10 noncommunicable disease (NCD) clinics across Malawi from April 2019 through December 2021. Eligible cohort participants were 18–65 years with depressive symptoms indicated by a PHQ-9 score ≥5. Questionnaires at the baseline and 3-month interviews included a vignette-based quantitative stigma instrument that measured treatment-related stigma, i.e., concerns about external stigma because of receiving depression treatment. Using inverse probability weighting to adjust for confounding and multiple imputation to account for missing data, this analysis relates participants' baseline levels of anticipated treatment stigma to the 3-month probability of achieving depression remission (i.e., PHQ-9 score < 5).

(NDA): https://nda.nih.gov/edit_collection.html?id=2822. The SHARP study data available from the NDA should be sufficient to replicate this paper's analyses. Data access requests may be submitted to the NDA in order to receive permission to access these data.

**Funding:** This work was supported by the National Institute of Mental Health, awarded to BWP (U19MH113202-01; https://www.nimh.nih.gov/). JMD was also supported by funding from the National Institute of Allergy and Infectious Diseases (T32AI070114; https://www.niaid.nih.gov/). The content of this manuscript is solely the responsibility of the authors and does not necessarily represent the official views of the National Institutes of Health or other funders. The funders had no role in study design, data collection and analysis, decision to publish, or preparation of the manuscript.

**Competing interests:** The authors have declared that no competing interests exist.

## Results

Of 743 included participants, 273 (37%) achieved depression remission by their 3-month interview. The probability of achieving depression remission at the 3-month interview among participants with high anticipated treatment stigma (0.31; 95% Confidence Interval [CI]: 0.23, 0.39)) was 10 percentage points lower than among the low/neutral stigma group (risk: 0.41; 95% CI: 0.36, 0.45; RD: -0.10; 95% CI: -0.19, -0.003).

## Conclusion

In Malawi, a reduction in anticipated depression treatment-related stigma among NCD patients initiating depression treatment could improve depression outcomes. Further investigation is necessary to understand the modes by which stigma can be successfully reduced to improve mental health outcomes and quality of life among people living with depression.

## Background

An estimated 13% of the global population live with a mental illness [1, 2], and 4% live with a depressive disorder [1, 2]. Amid effective, cost-efficient depression treatment options [3, 4] treatment gaps persist. Stigma is one barrier to treatment engagement that may exacerbate persistent treatment gaps [5–8]. Stigma is a multidimensional construct that is primarily defined by the "mark" of a discreditable attribute that subsequently reduces or devalues the personhood of people carrying that attribute [9, 10]. This phenomenon of stigma marking can vary based on the visibility of these attributes. Further, the degree to which these attributes are considered discreditable can also vary by social context and over time [11]. Stigmatizing experiences of individuals with mental illness can therefore vary based on the legibility of their symptoms and the discrediting value that such an illness possesses in their society.

Mental illness stigma is of global concern, yet much of the literature describing the manifestations of mental illness stigma to date has centered social norms in Europe and the United States. In sub-Saharan Africa, mental illness stigma research has focused on West Africa, while research in Southern Africa has been sparse. In Malawi, a Southeast African nation, published research on mental illness stigma is gradually expanding. For example, one study has described perceptions of mental illness in the nation's second-largest city, Blantyre; the results demonstrated that stigmatizing beliefs around mental illness vary in prevalence in this area compared to other international studies. For instance, participants attributed the cause of mental illness to God's punishment or other spiritual causes while also attributing mental illness to brain disorders [12]. Our study team has also contributed to the literature around depression stigma in Malawi specifically by validating an instrument to measure treatment-related, disclosure-related, and negative affect stigma within a population of patients with depression [13]. The scale-up of antidepressant treatment is a high priority for the Malawi Ministry of Health, and quantifiable information about the barriers to such a scale-up, such as stigma, is therefore relevant.

In previous mixed-methods research focused on patients who received depression treatment in Malawi, our study team found that some participants had concerns about treatment-related stigma when initiating depression treatment, and they expressed worry that this treatment-related stigma would influence their continued treatment engagement [14]. In Malawi, patients have also cited their support system, their ability to reach the clinic for appointments

(either due to material resources, transit access, or work/scheduling), and stressful life events (e.g., a dispute with a partner or family member) as key determinants to attending their care appointments [14–16]. Research in other social contexts has also demonstrated that suboptimal treatment engagement may reduce the effectiveness of depression treatment and increase time to depression remission [7]. Depression remission is a clinically important marker toward full depression recovery [17–19], and prior research indicates that patients with depression who receive clinically appropriate treatment can be expected to achieve remission by 3 months after treatment initiation [20]. Amid the scale-up of depression care in Malawi with sparse literature on stigma, the current study seeks to estimate the effect of patients' baseline anticipated treatment-related stigma on their 3-month probability of achieving depression remission after being newly identified with depression.

## Methods

### Parent study

The Sub-Saharan Africa Regional Partnership for Mental Health Capacity Building (SHARP) scale-up study is designed to compare the effect of two clinic-level implementation strategies on the successful integration of depression screening and treatment with other standard care at noncommunicable disease (NCD) clinics in Malawi. To that end, the study team monitored depression screening in 10 NCD clinics in Malawi from May 2019 to November 2021 and recruited eligible consenting participants into the patient cohort. Eligibility criteria required that participants be 18–65 years of age, have elevated depressive symptoms denoted by a score $\geq$5 on the Patient Health Questionnaire (PHQ-9) [21–24], and have a new or current diagnosis of diabetes or hypertension. These two chronic disease diagnoses were selected as part of the SHARP study eligibility criteria because depression often co-exists with hypertension and diabetes, which are rapidly increasing in prevalence in Malawi, and depression is associated with worse clinical outcomes for those comorbid conditions [25–27]. Participants were asked a series of questions, in Chichewa or Chitumbuka, at baseline, 3-month, 6-month, and 12-month follow-up interviews. During the baseline interview, research assistants re-assessed participants for depressive symptoms using the PHQ-9. Participants were excluded from this secondary analysis if they had PHQ-9 scores $\geq$5 at eligibility screening but not at the baseline interview (n = 203). The SHARP study protocol dictated that participants conduct their 3-month interview at exactly 3 months post-consent, but the protocol allowed these 3-month interviews to be scheduled within the bounds of 2 months and 5 months post-consent; 3-month interview data were considered missing if the 3-month interview was conducted outside of the window designated by the study protocol (n = 11). In this analysis, we related patients' baseline levels of anticipated depression treatment stigma to their probability of achieving depression remission by their 3-month follow-up interview.

### Exposure definition

The exposure of interest was high baseline treatment-related stigma (also referred to as treatment carryover), indicated by patients' responses to an instrument that our team has previously validated in this study population [13]. This instrument measures three domains of stigma: *negative affect*, or negative attitudes toward people with depression; *disclosure carryover*, or the stigma that participants expected would result from disclosing depression status; and *treatment carryover*, or participants' concerns that they would be treated as outsiders in their communities as a result of engaging in depression treatment [13, 28, 29]. The stigma instrument measured these domains using a vignette of a woman named Thandi and a series of eight prompts delivered on a 5-point Likert scale (see S1 Appendix for full prompts). Each

prompt was written such that agreement indicated endorsement of stigma. Strong agreement was equivalent to 4 points, while strong disagreement was 0 points. Responses were averaged per domain to produce three stigma summary scores per participant. For any domain, a neutral position was indicated by a score of 2 points. There were very few patients who maintained neutral positions on the treatment stigma domain. Thus, high anticipated treatment-related stigma was dichotomized as having a score greater than 2 in the treatment carryover domain. This dichotomy implies that participants in the high stigma group on average agreed with stigmatizing prompts in this domain, and participants not in that group generally disagreed or provided neutral responses to stigmatizing prompts. The two prompts included in the treatment carryover domain asked about whether the patient believed that Thandi (the vignette character) would lose friends if they knew that she was going to the clinic to 1) receive counseling or 2) receive medication for depression.

Our analysis centers anticipated treatment-related stigma for two reasons. First, because the patient population can largely be assumed to be treatment-naïve, the treatment carryover scale at baseline can be expected to capture participants' *anticipated* treatment-related stigma. By contrast, the wording of the other two scales' items is more likely to combine a mixture of participants' experienced stigma and anticipated stigma. Second, based on previous research, we expected that treatment-related concerns would have the most direct impact on treatment initiation and engagement and subsequently depression remission.

## Outcome definition

The primary outcome of interest is depression remission 3 months after study initiation. Depression remission is a binary outcome, defined as having a PHQ-9 score below 5 at the 3-month follow-up interview. Using this cut-off is supported by previous research on depression cut-off scores using the PHQ-9 [21–24]. Participants that answered positively to item 9 of the PHQ-9 (representing suicidal ideation) were still considered as remitted if their total PHQ-9 score was below 5. While suicidal ideation can be a symptom of depression, depression does not represent a necessary cause of suicidal ideation, and we therefore chose to allow flexibility for other etiologies that may also lead to suicidal ideation.

## Confounders

We used directed acyclic graphs (DAGs) to assist in identifying a sufficient set of confounders to estimate the causal relationship between anticipated treatment-related stigma and subsequent depression remission at the 3-month interview [30–33]. This sufficient set included baseline social support, baseline depressive symptoms, baseline wealth score, baseline adaptive coping behaviors, depression treatment assignment, education level, reported job at baseline, urbanicity, and stressful life events occurring in the three months preceding the baseline interview. While our DAG identified the relevance of patients' diabetes or hypertension morbidity to their depression treatment engagement and remission, it was nonetheless unrelated to patients' anticipated treatment-related stigma and therefore was not defined as a confounder. Social support was measured using the Multidimensional Scale of Perceived Social Support (MSPSS) [34, 35]. Stressful life events were measured using the Life Events Scale, and disjoint indicator variables were created based on the type of stressful life events experienced (see Table 1 for categories) [36, 37]. Adaptive coping behaviors were based on the Brief COPE scale; adaptive coping behaviors were believed to influence the way participants perceived treatment stigma as well as their ability to engage in depression treatment [38]. Treatment assignment is based on the depression treatment that patients were referred to by their clinicians at the NCD clinic at the time of the patients' initial depression screening and SHARP

study referral. Patients reported their education level and current employment during the baseline research interview and this information was encoded in statistical models using disjoint indicator variables. Urbanicity was a categorical variable derived from the NCD clinic that each patient attended: the clinics were classified as urban, peri-urban, or rural; in analyses, it was also encoded as a set of disjoint indicator variables. The wealth score was generated as a weighted factor score based on responses to radio, refrigerator, television, mobile phone, and car ownership, along with responses to whether the household had electricity, how often they worried about money, and how often they went to bed hungry [39, 40]. All scales were coded so that a larger number indicates a greater amount of that construct, e.g., a greater MSPSS score indicates greater social support. As a weighted factor score, the wealth score has a mean of 0 and standard deviation of 1 by design. With the exception of the life events scale, all other scales in this analysis were maintained as continuous variables. Further, to improve model fit, the following functional forms were selected: social support scores were modeled using a quadratic term, depressive symptom scores were modeled using restricted cubic splines with three knots, wealth scores used a quadratic term, and adaptive coping scores used linear splines with two knots.

## Missing data

A complete case analysis would exclude 69 (9%) participants due to missing data. Thus, to avoid loss in precision and possibly validity (assuming data missingness is not completely at random) [41–46], missing data were addressed using multiple imputation with chained equations. Data were assumed to be missing at random conditional on the exposure, outcome, and confounders described previously. In addition to these variables, the following partially observed variables were included in the imputation model: baseline health-related quality of life score (SF-8) [47, 48], baseline anxiety score (GAD-7) [49], baseline PTSD checklist (PCL-C) [50, 51], and stressful life events between baseline and 3-month follow-up. Fully observed variables included patient's PHQ-9 score from original eligibility screening, whether they were prescribed antidepressant medication after their initial screening, whether they were referred for supportive counseling after their initial screening, patient sex, patient education level, clinic, and patient age. Chained equations were conducted sequentially in order of least

Table 1. Summary of the analysis cohort (N = 743).

| Variable | Baseline | | | 3 Months | | |
|---|---|---|---|---|---|---|
| | Mean | SD | Missing | Mean | SD | Missing |
| Age (Years) | 50.8 | 9.9 | 0 | | | |
| Time Between Baseline and 3-Month Interview (Days) | | | | 100.1 | 17.7 | 48 |
| Depressive Symptoms (PHQ-9; scale range: 0–27) | 9.3 | 3.9 | 4 | 5.9 | 3.9 | 51 |
| Anxiety Symptoms (GAD-7; scale range: 0–21) | 7.1 | 4.2 | 2 | 4.6 | 3.8 | 49 |
| Stressful Life Events (LES; scale range: 0–14) | 3 | 2.3 | 0 | 2.4 | 2.2 | 47 |
| PTSD Symptoms (PCL-C; scale range: 17–85) | 42.9 | 14.6 | 1 | 35.2 | 13.6 | 49 |
| Social Support (MSPSS; 0–24) | 17.2 | 4.2 | 5 | 17.4 | 4 | 52 |
| Stigma Sub-Scale: Treatment Stigma (scale range; 0–4) | 1.4 | 1.1 | 1 | 1.2 | 1 | 52 |
| Stigma Sub-Scale: Negative Affect (scale range; 0–4) | 2.2 | 1 | 4 | 1.9 | 1 | 49 |
| Stigma Sub-Scale: Disclosure Carryover (scale range; 0–4) | 2.4 | 1.2 | 6 | 2.4 | 1.1 | 55 |
| Standardized SF8 Score[1] | 0.2 | 0.9 | 16 | 0.1 | 1 | 56 |
| Standardized Wealth Score[1] | 0 | 1 | 1 | | | |
| Adaptive Coping Behaviors (Brief COPE; scale range: 0–15) | 7.5 | 3 | 1 | 7.4 | 3.3 | 56 |

(*Continued*)

**Table 1.** (Continued)

| Variable | Baseline | | | 3 Months | | |
|---|---|---|---|---|---|---|
| | Mean | SD | Missing | Mean | SD | Missing |
| | N | % | | N | % | |
| **Sex** | | | | | | |
| Male | 158 | 21% | | | | |
| Female | 585 | 79% | | | | |
| **Education level** | | | | | | |
| No Formal School | 119 | 16% | | | | |
| Standard 1–5 | 238 | 32% | | | | |
| Standard 6–8 | 221 | 30% | | | | |
| Secondary School | 128 | 17% | | | | |
| Postsecondary School | 37 | 5% | | | | |
| **Employment** | | | | | | |
| Farmer | 328 | 44% | | | | |
| Business Owner | 131 | 18% | | | | |
| Homemaker | 142 | 19% | | | | |
| Other Employment | 92 | 12% | | | | |
| Not Currently Employed | 46 | 6% | | | | |
| *Missing* | 4 | 1% | | | | |
| **Urbanicity** | | | | | | |
| Rural | 139 | 19% | | | | |
| Peri-urban | 472 | 64% | | | | |
| Urban | 132 | 18% | | | | |
| **Baseline treatment referrals** | | | | | | |
| Only Antidepressant Medication | 116 | 16% | | | | |
| Only Counseling | 580 | 78% | | | | |
| Both Medication and Counseling | 38 | 5% | | | | |
| Neither | 9 | 1% | | | | |
| **High anticipated treatment-related stigma** | | | | | | |
| No | 563 | 76% | | 570 | 77% | |
| Yes | 179 | 24% | | 121 | 16% | |
| *Missing* | 1 | 0% | | 52 | 7% | |
| **Stressful Life Events—Baseline** | | | | | | |
| Employment Related | 128 | 17% | | 124 | 17% | |
| Safety Related | 40 | 5% | | 21 | 3% | |
| Personal Health Related | 181 | 24% | | 152 | 20% | |
| Romantic Relationship Related | 22 | 3% | | 12 | 2% | |
| Family Relationship Related | 214 | 29% | | 168 | 23% | |
| Death Or Illness of Loved One | 11 | 1% | | 7 | 1% | |
| *Missing* | 2 | 0% | | 52 | 7% | |
| **Depression remission (PHQ9<5) at 3-month interview** | | | | | | |
| No | | | | 418 | 56% | |
| Yes | | | | 273 | 37% | |
| *Missing* | | | | 52 | 7% | |

missing (n = 1) to most missing variable (n = 52). In total, 50 imputed data sets were generated with 500 iterations per imputation.

## Analytical approach

We estimated the marginal 3-month risk difference (RD) for depression remission contrasting high vs. low/neutral anticipated treatment-related stigma using inverse probability weighting. We estimated inverse probability of treatment (IPT) weights via a logistic regression model that predicted the probability of observing a positive exposure given a sufficient set of correctly specified confounding variables [52, 53]. Weights were assigned by predicting the probability of having high anticipated treatment stigma conditional on confounding variables (i.e., propensity score) and then separately predicting the crude probability of the high anticipated treatment stigma in each imputed data set, per participant. Stabilized weights for participants who were observed as having high anticipated stigma were applied by dividing the crude probability of having high anticipated stigma by the conditional probability. Likewise, stabilized weights for participants who were observed as having low/neutral anticipated stigma were applied by dividing the conditional probability of having low/neutral anticipated stigma by the crude probability of having low/neutral anticipated stigma. While the estimand of interest in this analysis was the average treatment effect using inverse probability of treatment weights, we used the same propensity scores to generate weights that would additionally estimate the average treatment effect in the treated (ATT) and untreated (ATU); we present all estimates in the main text (Table 2). Graphical and summary diagnostics were used to investigate misspecification of weights, and the diagnostics for these weights are also presented in the supplementary material (S2–S4 Tables, S1–S4 Figs).

After assessing the balance of covariates in the weighted population, we fit a weighted linear regression model that regressed depression remission at 3-month follow-up onto anticipated treatment-related stigma at baseline. Due to the computational complexity of combining multiple imputation with bootstrap methods, the confidence intervals were generated from multiple imputation alone and likely represent a conservative variance estimate [54–56]. All statistical analyses were conducted using Stata 16.1 [57].

## Ethics statement

This study was approved by the University of North Carolina Biomedical Institutional Review Board (UNC IRB) and the Malawi National Health Science Research Committee (NHSRC). Additional information regarding the ethical, cultural, and scientific considerations specific to inclusivity in global research is included in the Supporting Information (S1 Checklist).

## Results

In total, 946 participants enrolled in the SHARP cohort and completed their baseline interview; 743 (79%) participants had a PHQ-9 score $\geq$ 5 at the time of baseline re-assessment and

**Table 2. Estimates of standardized 3-month risk of depression remission, Risk Difference (RD), and 95% Confidence Intervals (CI) using inverse probability weighting (N = 743).**

| Estimand | Risk Group | Risk | 95% CI | RD | 95% CI |
|---|---|---|---|---|---|
| **Average Treatment Effect** | High Anticipated Treatment-Related Stigma | 0.31 | (0.23, 0.39) | -0.10 | (-0.19, -0.003) |
| | Low/Neutral Anticipated Treatment-Related Stigma | 0.41 | (0.36, 0.45) | 0. | |
| **Average Treatment Effect in the Treated** | High Anticipated Treatment-Related Stigma | 0.32 | (0.25, 0.39) | -0.07 | (-0.16, 0.04) |
| | Low/Neutral Anticipated Treatment-Related Stigma | 0.39 | (0.31, 0.46) | 0. | |
| **Average Treatment Effect in the Untreated** | High Anticipated Treatment-Related Stigma | 0.30 | (0.21, 0.40) | -0.11 | (-0.21, -0.01) |
| | Low/Neutral Anticipated Treatment-Related Stigma | 0.41 | (0.37, 0.45) | 0. | |

therefore met study inclusion criteria; 695 (95%) of these participants completed their 3-month follow-up interview within the appropriate window. Based on the imputation methods described previously, we were able to conduct analyses on the full sample of 743 participants. Table 1 represents the distribution of relevant variables prior to imputation; See S1 Table for the distribution of variables in the post-imputation data. The median number of days between the baseline interview and 3-month interview was 93 days (IQR:14 days). In general, this sample of participants had high levels of baseline social support (mean: 17.2, SD: 4.2) and post-traumatic stress symptoms (mean: 42.9; SD: 14.6; Table 1). The sample was also majority female (n = 585; 79%; Table 1), and the mean age was 51 years (SD: 9.9; Table 1). Most patients in this cohort (n = 618; 83%) were referred to counseling during their initial screening, and 154 (21%) were prescribed antidepressant medication during their initial screening. Only 9 (1%) patients were not referred to any treatment at initial screening. All 9 of these patients had PHQ-9 scores that would indicate treatment. Ultimately, 273 (37%) patients achieved depression remission by their 3-month interview (Table 1).

Patients' baseline treatment-related stigma scores indicated that most participants carried low or neutral levels of anticipated treatment-related stigma (n = 563; 76%; Table 1). On average, baseline treatment-related stigma scores tended to be lower than the scores for other stigma domains. For example, the average treatment-related stigma score was 1.4 (SD: 1.1), while the average disclosure carryover score was 2.4 (SD: 1.2; Table 1). Within the two treatment-related stigma prompts, 30% of patients (n = 223) agreed or strongly agreed that Thandi would lose friends if they found out that she was receiving counselling, and 27% (n = 200) agreed or strongly agreed that Thandi would lose friends if people found out that she took depression medication. Very few (≤20 patients) remained neutral when responding to each of these two prompts.

Finally, the IPT weighted estimate demonstrated that the risk of depression remission at the 3-month interview among participants with high anticipated treatment-related stigma was 0.31 (95% Confidence Interval [CI]: 0.23, 0.39; Table 2). Compared to the low/neutral treatment-related stigma group (risk: 0.41; 95% CI: 0.36, 0.45), the high treatment-related stigma group was 10 percentage points less likely to achieve depression remission by the 3-month interview (RD: -0.10; 95% CI: -0.19, -0.003; Table 2).

## Discussion

In this prospective analysis of anticipated treatment-related stigma and depression remission among people with hypertension or diabetes in Malawi, we saw that many participants (37%) achieved depression remission by their 3-month interview. Our analysis further demonstrated that achieving depression remission by the 3-month interview varied by anticipated depression treatment-related stigma, which was measured at the baseline interview. We estimated that the standardized risk (or probability) of remission at the 3-month interview was 41% among participants with low or neutral levels of anticipated treatment-related stigma, compared to 31% among the high anticipated treatment stigma group. The results suggest that, if we were to organize an intervention on treatment-related stigma that moved all participants from the high anticipated stigma group to the low/neutral group, we would expect to see a 10 percentage point increase in depression remission at the 3-month follow-up period in this population.

This study's findings align with other research globally that has demonstrated the detrimental effects of treatment stigma on continued treatment engagement and recovery from mental illness [5–8, 58–60]. Still, while several studies—the current analysis included—have identified stigma as an important barrier to public health efforts to improve access and engagement in clinically appropriate mental healthcare, there remains uncertainty on the most effective

means to meaningfully reduce stigma and improve mental health outcomes. Moreover, some research has shown that stigma reduction efforts tailored to the general public have failed at reducing key stigma targets, such as social distance toward individuals living with mental illness [61–64]. We believe that the current study contributes to this discussion by identifying anticipated treatment stigma among people with depression as a key domain that could be targeted for intervention to improve the probability of depression remission within the target 3-month window. While more research is warranted, it is possible that intervening on treatment-related stigma among people with depression and their immediate support network could increase their probability of depression remission by increasing support and reducing stigma around depression care-seeking. Indeed, in prior qualitative work our study team conducted among patients with perinatal depression, participants expressly identified the need for social support in coping with their depressive symptoms [15].

This study is not without its limitations. First, the treatment-related stigma scale focused on lost friendships, and a longer instrument may have been able to capture other features of treatment-related stigma concerns more fully. Second, as with all studies using latent variables to measure constructs, this study is vulnerable to measurement error. However, the latent variables used in this analysis were measured using validated instruments, providing further confidence that the variables represent their respective constructs [13, 14, 36–38, 51, 65, 66]. We anticipate that most measurement error occurred non-differentially, and we therefore expect that measurement error would more likely bias the current results toward a null treatment effect rather than a more extreme effect estimate [67]. Nevertheless, it is possible that participants modified their responses due to social desirability bias, and in such cases there may have been non-differential measurement bias in certain constructs used in this analysis; it would be difficult to predict the direction of such a bias [68], but we do not expect such a bias to have affected a large proportion of the participant sample. Finally, it is possible that a small number of participants (n ≤ 7) who were identified with depression and enrolled in the cohort had a history of depression treatment–in such a case, it could represent unmeasured confounding for this analysis. However, SHARP study data suggest that it is very rare (< 1%) for any participant to have a history of receiving depression care, and we therefore assumed that all patients in the sample were treatment naïve.

The study also has a number of strengths. First, we used a multidimensional stigma instrument, which allowed us to deconstruct the relationship between specific dimensions of stigma and other latent constructs. Second, the study had high participant retention, and the potential impact of missing data was addressed through multiple imputation assuming missingness to be at random conditional on measured variables. Third, the study used inverse probability weighting to estimate the total effect of anticipated treatment-related stigma on the 3-month probability of depression remission. Using a weighting approach in this analysis was important because it allowed us to estimate more policy-relevant effect estimates by reporting a risk difference rather than an odds ratio—while odds ratios are useful in many studies, our outcome of interest (depression remission) was far too prevalent for the odds ratio to approximate more easily interpretable measures such as the risk ratio [69].

Mental health stigma is ubiquitous and varies in degree by illness and social context, among other factors. In Malawi, a reduction in anticipated depression treatment-related stigma could improve depression remission outcomes. In this study, we found that patients with high anticipated treatment-related stigma had a 10-point reduction in their 3-month probability of depression remission. This suggests that patients who anticipate experiencing greater social isolation as a result of receiving depression treatment may not engage with treatment equally to those who do not have such heightened concerns. In order to improve depression outcomes for all patients with depression, it is necessary to address treatment-related stigma among

these patients and their immediate support networks. Further investigation is warranted to fully understand the modes by which such interventions on treatment-related stigma can be successfully reduced to improve mental health outcomes and quality of life among people living with depression.

## Supporting information

**S1 Checklist. Inclusivity in global research.**
(PDF)

**S1 Appendix. Stigma questionnaire.**
(PDF)

**S1 Table. Post-imputation distribution of analysis variables that required imputation (N = 743).** Data distributions reported in this table were drawn from the 1st imputed data set. Only imputed analytic variables (analytic variables that previously had missing observations) are displayed here.
(PDF)

**S2 Table. Pre- and post-weight distributions of variables used to design inverse probability of treatment weights (N = 743).** Data distributions reported in this table were drawn from the 1st imputed data set. Only variables used in the design of the inverse probability weights are displayed here.
(PDF)

**S3 Table. Post-weight distributions after applying inverse probability weights to estimate the average treatment effect in the treated[2] (N = 743).** Data distributions reported in this table were drawn from the 1st imputed data set. Only variables used in the design of the inverse probability weights are displayed here.
(PDF)

**S4 Table. Post-weight distributions after applying inverse probability weights to estimate the average treatment effect in the untreated[2] (N = 743).** Data distributions reported in this table were drawn from the 1st imputed data set. Only variables used in the design of the inverse probability weights are displayed here.
(PDF)

**S1 Fig. Pre- and post-weight distributions of baseline adaptive coping behaviors by treatment group (N = 743).** Data distributions reported in this table were drawn from the 1st imputed data set.
(TIF)

**S2 Fig. Pre- and post-weight distributions of baseline wealth score by treatment group (N = 743).** Data distributions reported in this table were drawn from the 1st imputed data set.
(TIF)

**S3 Fig. Pre- and post-weight distributions of baseline depressive symptoms by treatment group (N = 743).** Data distributions reported in this table were drawn from the 1st imputed data set.
(TIF)

**S4 Fig. Pre- and post-weight distributions of baseline social support by treatment group (N = 743).** Data distributions reported in this table were drawn from the 1st imputed data set.
(TIF)

## Author Contributions

**Conceptualization:** Josée M. Dussault, Melissa Stockton, Sherika Hill, Allison E. Aiello, Alexander Keil, Bradley N. Gaynes, Brian W. Pence.

**Data curation:** Chifundo Zimba, Harriet Akello, Melissa Stockton.

**Formal analysis:** Josée M. Dussault.

**Funding acquisition:** Bradley N. Gaynes, Michael Udedi.

**Investigation:** Chifundo Zimba.

**Methodology:** Josée M. Dussault.

**Project administration:** Chifundo Zimba, Harriet Akello, Bradley N. Gaynes, Michael Udedi, Brian W. Pence.

**Supervision:** Sherika Hill, Allison E. Aiello, Alexander Keil, Bradley N. Gaynes, Michael Udedi, Brian W. Pence.

**Writing – original draft:** Josée M. Dussault.

**Writing – review & editing:** Josée M. Dussault, Chifundo Zimba, Harriet Akello, Melissa Stockton, Sherika Hill, Allison E. Aiello, Alexander Keil, Bradley N. Gaynes, Michael Udedi, Brian W. Pence.

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
