## [Decision Letter · Decision Letter 0]

19 Dec 2022

PONE-D-22-27546Estimating the effect of anticipated depression treatment-related stigma on depression remission among people with depression in MalawiPLOS ONE

Dear Dr, Dussault,

Thank you for submitting your manuscript to PLOS ONE. After careful consideration, we feel that it has merit but does not fully meet PLOS ONE’s publication criteria as it currently stands. Therefore, we invite you to submit a revised version of the manuscript that addresses the points raised during the review process.

Please submit your revised manuscript by the Feb 02 2023 11:59PM. If you will need more time than this to complete your revisions, please reply to this message or contact the journal office at plosone@plos.org. Please include the following items when submitting your revised manuscript:A rebuttal letter that responds to each point raised by the academic editor and reviewer(s). You should upload this letter as a separate file labeled 'Response to Reviewers'.A marked-up copy of your manuscript that highlights changes made to the original version. You should upload this as a separate file labeled 'Revised Manuscript with Track Changes'.An unmarked version of your revised paper without tracked changes. You should upload this as a separate file labeled 'Manuscript'.

We look forward to receiving your revised manuscript.

Kind regards,

Dr Gayathri Delanerolle

Academic Editor

PLOS ONE

Reviewers' comments:

Reviewer's Responses to Questions

**Comments to the Author**

1. Is the manuscript technically sound, and do the data support the conclusions?

Reviewer #1: Yes

2. Has the statistical analysis been performed appropriately and rigorously? 

Reviewer #1: Yes

3. Have the authors made all data underlying the findings in their manuscript fully available?

Reviewer #1: Yes

4. Is the manuscript presented in an intelligible fashion and written in standard English?

Reviewer #1: Yes

5. Review Comments to the Author

Reviewer #1: This study explored the role of treatment-related stigma in depression care by estimating the effect of patients’ baseline anticipated treatment-related stigma on their 3-month probability of depression remission when newly identified with depression.

This work makes important contributions to the literature regarding the public mental health issues.

However, the manuscript can be further strengthened by some reframing of the introduction, addition of further details regarding the methods, and clarification of study findings. I have provided additional details below.

Since the participants in this study are all patients with chronic diseases, it is suggested that the author should make it clear in the title that this research group is NCD patients with depressive symptoms, rather than "people with depression" as currently described.

Abstract:

It should be clarified that this study is based on cohort study design.

In methods of abstract, the author stated that this study measured the data at four time points, but only analyzed the data after the third month in the text, which makes readers confused. I can understand that the data of this manuscript comes from a large research project and the observation at four time points is the design of the parent study. If the data in this manuscript does not use the follow-up data at the last two time points, it is recommended to modify this description in the abstract to avoid misunderstanding.

Introduction:

There are some descriptions of the limitations of existing literature around mental illness stigma. However, the rationale for studying this topic in Malawi specifically could be bolstered. Specifically, additional support is needed for why you explored a series of confounding factors in method section (P9, line 131-159)? The main confounding factors considered in this study should be justified in the background.

Methods:

Interesting that the author considered multiple psychosocial indicators as confounding factors, but did not mention participants' chronic diseases, such as NCD condition and other physical health related characteristics.You could link this more clearly to the limitations section which discusses the overall healthiness of your study population.

Please omit the following sentences, “In general, this sample of participants had high levels of baseline social support and post-traumatic stress symptoms (P12, line 28-219).” And instead describe the actual findings presented in Table 1.

Moreover, because the author did not describe the assessment method of social support and PTSD symptoms in method section, there is no basis for how to judge the "high level" here.

Result

The presentation of Table 2 is a bit confusing. It is suggested that the author present demographic characteristic variables together. Then it can be considered to put the observed variables after the baseline and three-month follow-up into a table and list them by two columns for comparison.

I think Table S3 can present the main results better than Table 3.

Discussion

As the author stated (in the methodology section), the stigma experienced by the participants was different from the anticipated stigma. However, I didn't see the author mention in the discussion what kind of targeted intervention should be adopted against the “anticipated treatment-related stigma”in this study. And, whether it is necessary to design and intervene the content of this type of stigma from the perspective of patients themselves, rather than reducing the discriminatory attitude towards community people in general? Please clarify.

6. PLOS authors have the option to publish the peer review history of their article (what does this mean?). If published, this will include your full peer review and any attached files.

Reviewer #1: No

---

## [Author Response · Author response to Decision Letter 0]

23 Jan 2023

We thank the reviewer for their thoughtful review. Please see the attached document "Response to Reviewers" for details on how we addressed reviewer feedback in this revised manuscript.

---

## [Editor Report · Decision Letter 1]

7 Feb 2023

Estimating the effect of anticipated depression treatment-related stigma on depression remission among people with noncommunicable diseases and depressive symptoms in Malawi

PONE-D-22-27546R1

Dear Dr Dussault,

We’re pleased to inform you that your manuscript has been judged scientifically suitable for publication and will be formally accepted for publication once it meets all outstanding technical requirements.

Kind regards,

Dr Gayathri Delanerolle

Academic Editor

PLOS ONE

---

## [Editor Report · Acceptance letter]

6 Mar 2023

PONE-D-22-27546R1 

Estimating the effect of anticipated depression treatment-related stigma on depression remission among people with noncommunicable diseases and depressive symptoms in Malawi 

Dear Dr. Dussault:

I'm pleased to inform you that your manuscript has been deemed suitable for publication in PLOS ONE. Congratulations! Your manuscript is now with our production department. 

Kind regards, 

on behalf of

Dr. Gayathri Delanerolle 

Academic Editor

PLOS ONE